# Canola Oil Ameliorates Obesity by Suppressing Lipogenesis and Reprogramming the Gut Microbiota in Mice via the AMPK Pathway

**DOI:** 10.3390/nu16193379

**Published:** 2024-10-04

**Authors:** Jing Gao, Li Ma, Jie Yin, Tiejun Li, Yulong Yin, Yongzhong Chen

**Affiliations:** 1Research Institute of Oil Tea Camellia, Hunan Academy of Forestry, Shao Shan South Road, No. 658, Changsha 410004, China; gaojing.he@163.com (J.G.); supermarry1@hnlky.cn (L.M.); 2National Engineering Research Center for Oil Tea Camellia, Changsha 410004, China; 3Yuelushan Laboratory, Changsha 410004, China; 4College of Animal Science and Technology, Hunan Co-Innovation Center of Animal Production Safety, Hunan Agricultural University, Changsha 410127, China; yinjie2014@126.com; 5Key Laboratory of Agro-Ecological Processes in Subtropical Region, Institute of Subtropical Agriculture, Hunan Provincial Key Laboratory of Animal Nutritional Physiology and Metabolic Process, Chinese Academy of Sciences, Changsha 410125, China; tjli@isa.ac.cn

**Keywords:** canola oil, obesity, lipid metabolism, gut microbiota, acetic acid, AMPK signaling pathway

## Abstract

Background: obesity is a worldwide problem that seriously endangers human health. Canola oil (Col) has been reported to regulate hepatic steatosis by influencing oxidative stress and lipid metabolism in Kunming mice. However, whether Col exhibits an anti-obesity effect by altering the gut microbiota remains unknown. Methods: in this study, we observed that a high-fat diet increased lipogenesis and gut microbiota disorder in C57BL/6J male mice, while the administration of Col suppressed lipogenesis and improved gut microbiota disorder. Results: the results show that Col markedly reduced the final body weight and subcutaneous adipose tissue of C57BL/6J male mice fed a high-fat diet (HFD) after 6 weeks of administration. However, although Col did not effectively increase the serum concentration of HDL, we found that treatment with Col notably inhibited the low-density lipoprotein (LDL), total cholesterol (TC), and triglycerides (TGs) in HFD mice. Furthermore, Col ameliorated obesity in the liver compared to mice that were only fed a high-fat diet. We also found that Col significantly inhibited the relative expression of sterol regulatory element binding protein (SREBP1/2), peroxisome proliferator-activated receptor γ (PPARγ), and insulin-induced genes (Insig1/2) that proved to be closely associated with lipogenesis in HFD mice. In addition, the concentration of acetic acid was significantly increased in Col-treatment HFD mice. Further, we noted that Col contributed to the reprogramming of the intestinal microbiota. The relative abundances of *Akkermansia*, *Dubosiella*, and *Alistipes* were enhanced under treatment with Col in HFD mice. The results also imply that Col markedly elevated the phosphorylation level of the AMP-activated protein kinase (AMPK) pathway in HFD mice. Conclusions: the results of our study show that Col ameliorates obesity and suppresses lipogenesis in HFD mice. The underlying mechanisms are possibly associated with the reprogramming of the gut microbiota, in particular, the acetic acid-mediated increased expression of *Alistipes* via the AMPK signaling pathway.

## 1. Introduction

The rate of obesity is rising with time around the world, and there are few efficient management strategies due to its complex pathogenesis [1]. Excessive body weight and fat in the abdomen and viscera are harmful to health and are generally recognized as signs of obesity. Multiple studies have shown that the degree of obesity has a positive relationship with the mortality/morbidity risk with respect to complications such as inflammation, non-alcoholic fatty liver disease, type II diabetes, and cardiovascular diseases [2]. Decreasing the incidence of obesity and its related diseases is an imminent issue. Many studies have attempted to determine the relationship between obesity and energy balance, but nothing has been demonstrated to curb the obesity epidemic, and every country has been suffering from this crisis [3].

The types and quantities of cooking oils consumed daily have been considered as a key influence on the causes of obesity. Firstly, oil is closely associated with lipid metabolism in the body. Secondly, daily cooking oil intake contributes to reprogramming the biochemical composition of the gut microbiota, which is influenced by high-fat diet-induced disturbances of the gut microbiota in mice [4]. Our studies have demonstrated the potential mechanisms of the physiological effects exerted by Col in HFD-induced lipid dysmetabolism, which may be associated with reprogramming of the gut’s microbial functions, especially *Alistipes*-mediated acetic acid production.

Col is a vegetable oil derived from the seeds of the rapeseed plant, which belongs to the Brassicaceae family [5]. Col is low in saturated fatty acids (SFAs) and rich in monounsaturated fatty acids (MUFAs) and polyunsaturated fatty acids (PUFAs), particularly alpha-linolenic acid (ALA, omega-3 fatty acid) [6]. Consuming oils high in saturated fats can contribute to an imbalance in lipid metabolism and lead to weight gain. However, Col’s composition is more beneficial for weight management [7]. Additionally, the ALA in Col has potential anti-obesity effects, including reducing body weight, fat mass, and inflammation. ALA has also been linked to improved insulin sensitivity, which is important for regulating blood sugar levels and preventing weight gain [8]. Studies have shown that consuming Col can lead to a reduction in LDL and triglyceride levels, with an increase in HDL cholesterol, which is conducive to lipid metabolism in the liver. Furthermore, Col contains plant sterols, which are structurally similar to cholesterol and can compete with cholesterol for absorption in the gut. This leads to a decrease in cholesterol absorption and an overall reduction in blood cholesterol levels [9]. Overall, the fatty acid composition and other components of Col, such as vitamin E, plant sterols, tocopherols, and other antioxidants, make it a good choice for maintaining healthy lipid metabolism, which can help combat obesity and reduce the risk of cardiovascular disease [10].

The gut microbiota is strongly shaped by HFD. Studies have indicated that gut composition is closely associated with obesity, based on differing gut microbiota compositions between lean and obese mice [2]. Interestingly, the gut microbiota can be sensitive to different dietary oils, but how Col regulates the composition of the gut microbiota and exhibits anti-obesity effects is still unclear. In addition, the literature indicates that metabolites of gut microbes have a strong relationship with short-chain fatty acids (SCFAs). The microbiota in the gut, such as *Streptococcus*, *Acetobacteraceae*, *Precottella*, *Clostriudium*, *Bifidobacterium*, and *Bacteroides*, usually promote the production of acetic acid, while *Bacteroides* and *Akkermansia* contribute to the synthesis of propionate [11]. In addition, several varieties of *Faecelibacterium* and *Rumincoccus* have a tendency to increase the production of butyrate. In turn, the types of SCFAs can influence the composition of the gut microbiota. Importantly, alterations in both the gut microbiota and SCFAs are responsible for lipid metabolism [12]. 

In this study, we investigated the effect of Col on the lipid metabolism and gut microbiota in mice. Eventually, Col inhibited HDF-induced obesity and suppressed lipogenesis. The underlying mechanism was suggested to be the reordering of the gut microbiota, in particular, the acetic acid-mediated increased expression of *Alistipes* via the AMPK signaling pathway.

## 2. Materials and Methods 

### 2.1. Analysis of the Fatty Acid Profile of Col

Col (relative content higher than 93%) was purchased from a supermarket chain (Jinlongyu ®, RT-MART, Changsha, China). Oil was mixed with an extracting solution, followed by evaporation in a vacuum concentrator, adding Methanol and Trimethylsilyl diazomethane, vortex mixing for 10 s, and standing at room temperature for 15 min. Nitrogen was used for drying, followed by adding n-Hexane and redissolving. Centrifugation was undertaken for 5 min at 12,000 rpm and 4 °C. We then transferred the supernatant into a fresh vial for GC-MS analysis. The detection method of GC-MS analysis was described in our previous article [13]. To determine the fatty acid composition of Col, 49 types of free fatty acid substances were measured. We found that Col consists of monounsaturated fatty acids (72.4%), polyunsaturated fatty acids (20.51%), and a small amount of saturated fatty acids (7.04%) (Table 1).

### 2.2. Animals and Diets 

The experiment proceeded in compliance with the Chinese guidelines for animal welfare. The experimental protocol was approved by the Animal Care and Use Committee of the Chinese Academy of Sciences, and the ethical approval code is ISA2017030523. C57BL/6J mice (male, 18.095 ± 0.16 g) from SLAC Laboratory Animal Central (Changsha, China) were chosen in this study and randomly distributed into two groups (10 replicates for one treatment, 20 replicates for another treatment). Mice with 20 replicates were fed a high-fat diet during the trial during the first 6 weeks to induce obesity, while the 10-replicate group received the control diet with a lower fat content. Twenty replicate mice that were fed the HF diet were randomly divided into two groups (high-fat diet group (HFD) and Col group (Col)), and the remaining mice comprised the control group (Cont). 

All mice were given unrestricted access to food and drinking water and were raised in a room at 25 °C, with 45% to 60% humidity and a 12 h/day lighting cycle. The diets used in this experiment were as reported in our previous study [14]. 

### 2.3. Col Treatment 

During the gavage experiment period, the Cont and HFD groups were fed the basal diet and high-fat food, respectively. Each mouse was given a saline gavage at 8:00 am every morning, while Col group mice were given a high-fat diet and a 9 g kg^−1^ d^−1^ dose of Col at the same time to identify the influence of Col on lipid metabolism in obese mice. According to our earlier findings, normal C57BL/6J male mice’s growth performance was greatly enhanced by 9 g kg^−1^ d^−1^ as opposed to 6 g kg^−1^ d^−1^ and 12 g kg^−1^ d^−1^ by Col gavage. After 6 weeks of Col gavage treatment, all mice were euthanized. Before the final day of the trial, all mice fasted overnight and were euthanized via an intravenous injection of sodium pentobarbital (50 mg/kg BW, Sigma, St. Louis, MO, USA) on the last day of the trial. Then, serum, liver, adipose tissue, and colonic digesta samples were collected and weighed.

### 2.4. Biochemical Index Parameters 

We collected serum from blood samples with centrifugation under the following conditions: 4 °C, 3000× *g*, 10 min. We then chose appropriate commercially available porcine-specific kits to determine the concentration of high-density lipoprotein (HDL), low-density lipoprotein (LDL), triglycerides (TGs), total cholesterol (TC), and glucose (Glu) utilizing a Cobas c-311 Colter Chemistry Analyzer.

### 2.5. Protease Activity Assay

Separated liver samples were used to test lipase activity, including hepatic lipase, lipoprotein lipase, lipases, and lipase synthase, as these indexes are commonly dysregulated in HFD-fed or obese subjects. An assay kit (Huamei Biological Engineering firm, Wuhan, China) was used to measure all lipase activity. Another assay kit (Nanjing Jiancheng Institute of Biological Engineering, Nanjing, China) was used to evaluate the activity of acetyl-CoA carboxylase.

### 2.6. Reverse-Transcription PCR 

Unfrozen liver samples (preserved in liquid nitrogen) were used to isolate total RNA via TRIzol reagent (Invitrogen, Waltham, MA, USA). We added DNase I (Invitrogen, Waltham, MA, USA) to the total RNA. β-actin served as an internal reference for the processing of the signals and data that were obtained. Reverse transcription was conducted at 95 °C for 5 s, while PCR cycling was conducted for 30 s at 58 °C, then 20 s at 72 °C. The relative expression of genes chosen in the trial was calculated and expressed as a ratio to the expression in the control group (Table 2). 

### 2.7. Short-Chain Fatty Acids (SCFAs) 

Colonic content was diluted with water to appropriate concentrations and stored at −80 °C to determine the concentration of SCFAs, including propionic acid, acetic acid, isobutyric acid, butyrate, isovaleric acid, and pentanoic acid. Metabolite extraction methods were used according to the chemical characteristics of multi-targeted metabolites via capillary column gas chromatography (GC-14B, Shimadzu, Kyoto, Japan; capillary column: 30 m × 0.32 mm × 0.25 μm film thickness).

### 2.8. Gut Microbiota

We labeled primers with a barcode (16S V3-V4) upon obtaining the fecal samples’ whole DNA. Purified, quantified, and homogenized PCR products were collected to generate and analyze sequencing libraries (according to our previous study). Operational taxonomic units (OTUs) with a 97% match were clustered to determine the different species composition diversity of the samples using Tax4Fun analysis and a *t*-test between groups was performed (*p* < 0.05) to determine the significant difference between groups. 

### 2.9. Statistical Analysis 

One-way analysis of variance (ANOVA) was used for all the statistical analyses in this study. Multiple comparisons were carried out via Bonferroni analysis. Additionally, a Spearman correlation study was performed. The means ± standard errors of the means (SEMs) were used to express the data. *p* < 0.05 was used as the statistical significance threshold. GraphPad Prism 9 was used to construct all figures.

## 3. Results 

### 3.1. Col Inhibits the Body Weight Gain of Obese Mice 

The body weight change and weight gain in both trials were monitored after euthanizing the mice. As anticipated, the mice that underwent Col intervention showed a lower body weight change and weight growth compared to the HF-fed mice under normal saline treatment (Figure 1A–D) (*p* < 0.001).

### 3.2. Col Inhibits the Subcutaneous Adipose Accumulation of Obese Mice 

The weight of subcutaneous adipose tissue and liver weights were tested to examine the effect of Col on lipid metabolism. Col treatment reduced the ratio of subcutaneous adipose tissue/body weight (*p* <0.01) but had no effect on the liver weight (ns) (Figure 2A,B). 

### 3.3. Col Inhibits the Biochemistry Indexes of the Lipids of Obese Mice 

We measured the levels of HDL, LDL, TC, TG, and Glu in mice to determine the Col regulatory effect on serum and the liver. As compared to the HFD group, Col significantly decreased the serum levels of LDL (*p* < 0.01), TC (*p* < 0.05), TG (*p* < 0.001), and Glu (*p* < 0.001), while no difference was observed in serum HDL levels between the groups (*p* > 0.05) (Figure 3A–E). Interestingly, the results show that the Col treatment significantly restored the disordered liver biochemical indexes induced by a high-fat diet to normal levels, except for the concentration of Glu, such as HDL (*p* < 0.001), LDL (*p* < 0.001), TC (*p* < 0.001), and TG (*p* < 0.001) (Figure 3F–J). All these findings suggest that Col fully restores dyslipidemia in obese mice.

### 3.4. Col Regulates the Hepatic Lipase Bioactivity of Obese Mice 

The results show that the activity of multiple hepatic lipases is closely associated with the concentration of circulating high-density lipoproteins and is easily influenced by dietary intake [15]. Thus, we calculated the concentrations of lipases, hepatic lipase, lipoprotein lipase, lipase synthase, and acetyl-CoA carboxylase. According to the data, Col intervention lasting 6 weeks reduced the activity of the hepatic lipases induced by the high-fat diet to normal levels, especially the concentration of acetyl-CoA carboxylase (*p* < 0.001) (Figure 4A–E). 

### 3.5. Col Inhibits the Hepatic Lipid Anabolism of Obese Mice 

As we found that the Col treatment significantly influenced subcutaneous adipose accumulation and hepatic lipase activity, we detected the gene expression related to lipid metabolism to determine the underlying mechanism. We first determined the expression of genes that regulate lipid anabolism. Notably, Col treatment significantly decreased both mRNA and protein expression, including SREBP1 (*p* < 0.001), SREBP2 (*p* < 0.01), Insig1 (*p* < 0.001), Insig2 (*p* < 0.001), PPAR (*p* < 0.01), and PPARγ (*p* < 0.01), compared to the HFD group, except for acetyl-CoA carboxylase (ACC) (*p* > 0.05). Secondly, we determined the relative expressions of Liver X receptor α (LXRα) and Liver X receptor β (LXRβ) that have been reported to be closely related to lipid catabolism. The results indicate that Col treatment had a significant influence on the protein expression of LXRα (*p* < 0.01) compared to the HFD mice but failed to regulate the mRNA and protein relative expression of LXRβ (*p* >0.05). Col treatment made a significant contribution to ameliorating obesity induced by a high-fat diet, which may be closely associated with the inhibition of hepatic lipid anabolism (Figure 5A–P).

### 3.6. Col Inhibits the AMPK Pathway Activity of Obese Mice

By assaying the activity of the AMPK signaling pathway, we found that Col suppressed the expression of the AMPK pathway that was promoted by a high-fat diet compared to obese mice (*p* < 0.05) (Figure 6).

### 3.7. Col Reorganizes the Gut Microbiota Communities of Obese Mice

According to previous reports, HFD treatment can alter the gut microbial composition, which can induce disturbances in mouse and human metabolism. To verify the relationship between Col treatment and obesity, we extracted fecal DNA and used 16S rRNA gene sequencing to examine the gut microbiota. After deleting the low-quality sequences, 58,660 clear reads were produced out of the 60,111 average raw reads that were found. In comparison to HFD mice, Col treatment elevated the variety of the gut microbiota, with the exception of the Shannon, Simpson, chao1, and ACE measures, as demonstrated by the alpha diversity of the gut microbiota. More significantly, in comparison to the HFD group, Col raised the detected species, chao1, and ACE indexes, while Shannon and Simpson’s indexes fell short of the normal values. In summary, Col treatment restored the alpha diversity to normal levels compared to HFD mice (Figure 7A).

To determine the exact relationship between Col regulation and the gut microbiota community in the obesity model, we observed the percentages in the top tens at phylum, order, and genus levels. At the phylum level, *Firmicutes*, *Bacteroidota*, *Verrucomicrobiota*, *Proteobacteria*, *Campylobacterota*, *Actinobacteriota*, *Deferribacteres*, *Desulfobacterota*, and *Spirochaetota* formed the ten largest microbial constituents in each group. We implemented a statistical examination of the majority of gut microbiota. Overall, gavage treatment with Col restored the gut microbial composition of obese mice more closely to that of normal mice. For instance, Col significantly increased the abundance of *Firmicutes*, *Proteobacteria*, and *Desulfobacterota* compared to the HF group, while no difference in the three-phylum microbiota was observed between the Cont and Col mice. In addition, Col enriched the composition of *Bacteroidota*, *Verrucomicrobiota*, *Campylobacterota*, *Actinobacteriota*, *Deferribacteres*, and *Spirochaetota* in obese mice compared to mice that did not receive the Col intervention (Figure 7B). Likewise, *Bacteroidales*, *Verrucomicrobiales*, *Erysipelotrichales*, *Staphylococcales*, *Bacillalesm*, *Lachnospirales*, *Bacteria*, *Erysipelotrichales*, *Saccharimonadales*, *Rhizobiales*, and *Desulfovibrionales* comprise the top-ten species at order level. Interestingly, the trends in interventions with respect to Col in obese mice were consistent with the variety at the phylum level, and Col enhanced the relative abundances of *Erysipelotrichales*, *Staphylococcales*, and *Bacillalesm* to those of regular mice compared to the HFD group (Figure 7C). Meanwhile, the results show that at the genus level, *Akkermansia*, *Staphylococcus*, *Dubosiella*, *Lysinibacillus*, *Lachnoclostridium*, *Saccharimonas*, *Lachnospiraceae*, *Alistipes*, *Helicobacter*, and *Alloprevotella* comprised the ten largest microbial abundances in mice. Apparently, Col closely influenced the relative composition of *Dubosiella*, *Alistipes*, and *Alloprevotella*; significantly increased *Dubosiella*, *Alistipes*, and *Alloprevotella* compared to HFD mice; and no alteration was observed in the Cont and Col mice (Figure 7D). 

In general, Col can reverse disorders with respect to the composition and relative abundance of gut microbiota in obese mice. 

### 3.8. Col May Exhibit Anti-Obesity Roles via Microbiota-Mediated Acetic Acid in Obese Mice

HFD modifies the synthesis and conversion of metabolites linked to the gut microbiota, such as bile acids and SCFAs, which are crucial for controlling lipid absorption and metabolism. After detecting the concentration of SCFAs, we observed that the HFD raised the concentrations of butyric acid, isobutyric acid, isovaleric acid, and pentanoic acid, and Col gavage treatment realigned the variations. Col significantly enhanced the content of acetic acid compared to the other groups (*p* < 0.01) and significantly inhibited the concentration of valeric acid compared to HFD mice (*p* < 0.01). Notably, although no significant difference was observed in the variation of fecal propionic acid, butyric acid, isobutyric acid, and isovaleric acid, Col had a tendency to regulate this occurrence to a normal degree (Figure 8).

Alternatively, we supposed that the altered microbiota produced different levels of acetic acid. Notably, in this study, we observed altered *Dubosiella*, *Alistipes*, and *Akkermansia*, all of which are genera that produce acetic acid. As a result, Pearson correlation analysis was used to analyze the relationships between acetic acid and *Akkermansia*, *Dubosiella*, and *Alistipes*. Remarkably, the relative abundances of *Alistipes* showed a strong correlation with the production of acetic acid in feces (*p* <0.05). This suggests that *Alistipes* was primarily responsible for the synthesis of acetic acid in this study. All of these findings suggest that Col therapy enhanced lipid metabolism in mice given the HFD by modifying the composition of the gut microbiota, particularly *Alistipes*, which increased the production of acetic acid (Figure 9). 

## 4. Discussion

Canola oil (Col) is an edible oil with high nutritional value, which is rich in vitamin E, carotene, saturated and unsaturated fatty acids, phospholipids, sterols, stigmasterol, squalene, and other nutrients [16]. The results of previous studies have suggested that women with lipid disorders may benefit from consuming canola oil rather than sunflower oil [17]. Moreover, canola oil may decrease LDL cholesterol compared to olive oil, sunflower oil, and sources of SFA, and may also reduce body weight compared to other oils [18]. Furthermore, certain seed phospholipids found in Col are crucial for the growth of blood vessels, nerves, and the brain [19]. In this trial, 6 weeks of Col gavage treatment reduced the serum and hepatic concentrations of LDL, TG, and Glu in HFD mice. However, there have been few investigations on the precise impact of Col on obesity-related to SCFAs. Consequently, we designed our experiment to determine the regulatory ability of Col on obesity by researching the connection between SCFAs and the gut microbiota.

The AMPK signaling pathway is known as a key impactor in cardiovascular diseases and various liver diseases; the exact mechanism may touch on the regulatory effects on lipid metabolism [20]. In animals with alcohol- or insulin-induced fatty liver disease, type 2 diabetes, and obesity, upregulation of AMPK can mitigate the condition [21]. Additionally, the majority of naturally occurring AMPK agonists can control hepatocyte oxidative stress and lipid metabolism, all of which, in turn, can control fatty liver disease in mice [22]. In particular, we observed that Col treatment could downregulate the activity of the AMPK signaling pathway. In this trial, the AMPK signaling pathway significantly increased the activity of hepatic lipases after treatment with Col in the HFD group. SREBP2 is directly phosphorylated by AMPK, which decreases SREBP2’s nuclear translocation and reduces HMGCR transcription [23]. Moreover, medications that activate insig1, which targets AMPK, lower active SREBP levels and, as a result, lower target gene expression, which includes TG synthesis genes [24]. All of these findings show that AMPK activation and SREBP2 suppression may result in lower LDL levels [25]. 

In adipose tissue, the target genes of PPARγ are mainly related to fat synthesis and transformation, and SREBP1 forms a sophisticated regulatory network with PPARγ signaling molecules, which is one of the major pathways for the regulation of lipid metabolism [26]. Studies have shown that the expression of the SREBP1 gene is involved in the accumulation of intramuscular fat in the muscle of lactating sows and plays an important role in regulating the deposition of muscle fat after delivery [27]. SREBP1 can be ectopically expressed in 3T3-L1 and HepG2 cells, suggesting that SREBP1 induces the level of endogenous PPARγ mRNA and enhances the adipogenic differentiation of adipose tissue [28]. Dietary addition of soybean oil has been shown to increase the mRNA level of PPARγ in skeletal muscle and significantly increase intramuscular fat in fattening pigs, as well as significantly inhibiting or preventing the occurrence of adipogenic differentiation when the expression of PPARγ is knocked down or completely knocked out [29]. Both SREBP1 and PPARγ genes are located downstream of the AMPK signaling pathway, and their transcriptional activities are regulated by AMPK, which is the main upstream effector of energy- and insulin-mediated regulation of SREBP1 expression [30]. Activation of the AMPK signaling pathway significantly up-regulates the expression of the SREBP1 gene and thus activates the synthesis of lipids [31]. In addition, when the AMPK signaling pathway is activated, 4EBP1 can be phosphorylated to increase the expression of PPARγ, which can promote fat synthesis and adipocyte differentiation, regulate lipid metabolism in pigs, and further participate in fat deposition and good meat quality [32]. The AMPK signaling pathway regulates the expression of SREBP1 and PPARγ, which plays a decisive role in adipocyte differentiation [33]. In this study, suppressing the AMPK pathway signal inhibited the expression of Insig1/2 and SREBP1/2 as well as PPAR/γ under the intervention of Col in HFD mice. 

Obesity manifests as the excessive absorption and buildup of fat. The main tissues responsible for storing lipids are the liver and white adipose tissue [34]. White adipose tissue and non-alcoholic fatty liver frequently undergo excessive enlargement as a result of excessive lipid accumulation [35]. Mixed triglycerides constitute the main component of the diet. The human body has to hydrolyze exogenous fat for it to be absorbed. Lipases, hepatic lipase, lipoprotein lipase, acetyl-CoA carboxylase, and lipase synthase are the main lipases found in the digestive tract [36]. The direct raw material for fatty acid synthesis is acetyl CoA, which is derived from the catabolism of sugars, ketone bodies, and proteins [37]. Subsequently, acetyl-CoA is converted to malonyl-CoA by acetyl-CoA carboxylase, and the nascent fatty acid chain is lengthened by fatty acid synthase (FASN) to synthesize products such as palmitic acid [38]. During this process, acetyl-CoA carboxylase and FASN are directly regulated by SREBP [39]. In addition, the activity of acetyl-CoA carboxylase is regulated by AMPK, and nicotinamide adenine dinucleotide phosphate is often consumed during fatty acid synthesis, which triggers AMPK to inhibit acetyl-CoA carboxylase via phosphorylation, thus blocking fatty acid synthesis [40]. Further, the product of acetyl-CoA carboxylase, malonyl coenzyme A, is reduced, the inhibition of fatty acid oxidation rate-limiting enzyme is weakened, and fatty acid oxidation metabolism is enhanced [41]. More importantly, PPARα is a key transcription factor regulating the metabolism of fatty acid oxidation [42]. Consistent with previous studies, we indicated that Col treatment significantly raises the activity of acetyl-CoA carboxylase, resulting in enhanced fatty acid oxidation, thereby inhibiting the obesity induced by a high-fat diet. The exact relationship between the anti-obesity effects of Col treatment and acetyl-CoA carboxylase may be associated with the suppressed expression of SREBP, PPAR, and the AMPK pathway.

Several recent studies have demonstrated the importance of the intestinal microbiota and its metabolites on human health [43]. SCFAs generated by the gut microbiota have a wide range of biological effects on the host, such as anti-obesity, anti-diabetes, anti-cancer, and anti-inflammatory [44] effects. Acetate, propionate, and butyrate are the three primary SCFAs produced by gut bacteria, accounting for 90% of the total [45]. As stated previously, SCFAs play a crucial role in obesity by influencing appetite [46]. For example, when evaluating SCFA fecal levels in obese, overweight, and lean individuals, obese participants produce the largest amounts of SCFAs [47]. In addition, an investigation found that the process of SCFAs lowering dietary intake and fat storage was associated with vagal afferents via regulating key genes and hormones, including mitochondrial transcription factor A, free fatty acid receptor 2, and cytochrome-C oxidase IV [48]. In this study, Col significantly enhanced the content of acetic acid compared to other groups, which strongly supports these opinions. Moreover, SCFAs could inhibit the lipid synthesis process by suppressing hepatic acetyl-CoA carboxylase activity and regulating the related expression of PPARα [49]. In the current study, Col significantly inhibited the relative expression of SREBP1/2 and PPARγ, which proved to be closely associated with lipogenesis in HFD mice. We also indicated that Col treatment increased the activity of acetyl-CoA carboxylase. Thus, this study implies that Col acts as a key impactor in the regulation of obesity.

Microorganisms precisely carry related lipoproteins from the gastrointestinal tract to the liver, regulating lipid metabolism and helping to moderate aberrant blood cholesterol levels [50]. Endogenous elements or nutrients obtained from the diet have the greatest influence on lipid metabolism. Growing evidence suggests that the composition of the gut bacteria influences the body’s lipid levels [51]. In obese animal models, the gut bacteria composition is drastically altered. However, there is no convincing proof of Col’s positive anti-obesity effects on gut microbes. During this experiment, the results indicated that Col ameliorated obesity and suppressed lipogenesis in HFD mice through reprogramming of the gut microbiota via the acetic acid content. Initial observations suggest that dyslipidemia leads to reduced diversity of bacteria in the gut microbes, which is similar to our findings [13]. In comparison to the HFD group, Col increased the detected species, chao1, and the ACE index, while the Shannon and Simpson indexes fell short of normal levels. In summary, we believe Col has a positive impact on obesity-related gut microbiota disorders. 

Col treatment significantly influenced the balance of the gut microbiota in HFD mice in this research. We found that Col has an effect on numerous microbiota components, including the abundance of *Firmicutes*, *Proteobacteria*, and *Desulfobacterota*. Consistent with previous studies, Col increases the relative abundance of *Desulfobacterota.* Interestingly, Col decreases the contents of *Firmicutes* and *Proteobacteria*, which is contrary to previous results. The ratio of *Firmicutes* to *Bacteroidota* was reported to be inversely proportional to obesity, and the results demonstrate this point. In addition, Col enriched the composition of *Bacteroidota*, *Verrucomicrobiota*, *Campylobacterota*, *Actinobacteriota*, *Deferribacteres*, and *Spirochaetota* in obese mice compared to mice that did not receive the Col intervention. Correspondingly, Col enhanced the relative abundances of *Erysipelotrichales*, *Staphylococcales*, and *Bacillalesm* in regular mice compared to the HFD group at the phylum level, while Col significantly increased *Dubosiella*, *Alistipes*, and *Alloprevotella* compared to HFD mice at the genus level. Early reports show that *Dubosiella* could increase due to a high-fat diet proportionally to obesity but inversely proportionally to SCFAs, and all data are consistent with these results. Moreover, *Alistipes* are SCFA producers, which could alleviate obesity through the modulation of genes related to lipid metabolism. Accordingly, the relative abundances of *Dubosiella* and *Alistipes* were positively associated with Col in this research. 

To determine the associations between *Alistipes* and SCFAs with respect to the underlying regulation effect of Col on obesity, we conducted a correlation analysis between the gut microbiota and acetate, propionate, and butyrate. Remarkably, the relative abundances of *Alistipes* showed a strong correlation with the production of acetic acid in feces. This suggests that *Alistipes* was primarily responsible for the synthesis of acetic acid in this study and also suggests the anti-obesity efficacy of Col in HFD mice. In summary, obese mice exhibit a disordered gut microbiota and SCFA metabolites. All of these findings suggest that Col therapy enhances lipid metabolism in mice given an HFD by modifying the composition of the gut microbiota, particularly *Alistipes*, which increases the production of acetic acid, thus reducing obesity.

## 5. Conclusions

Col has an anti-obesity effect and alleviates lipogenesis through increasing acetyl-CoA carboxylase bioactivity, suppressing the expression of SREBP1/2, PPARγ, and sig1/2, enhancing the relative abundances of *Dubosiella*, *Alistipes*, and *Alloprevotella*; enriching the acetic acid contents; and inhibiting AMPK phosphorylation. The results of this study suggest that Col ameliorates obesity and suppresses lipogenesis in HFD mice. The underlying mechanisms are possibly associated with the reprogramming of the gut microbiota, in particular, the acetic acid-mediated increased expression of *Alistipes* via the AMPK signaling pathway (Figure 10).

## Figures and Tables

**Figure 1 nutrients-16-03379-f001:**
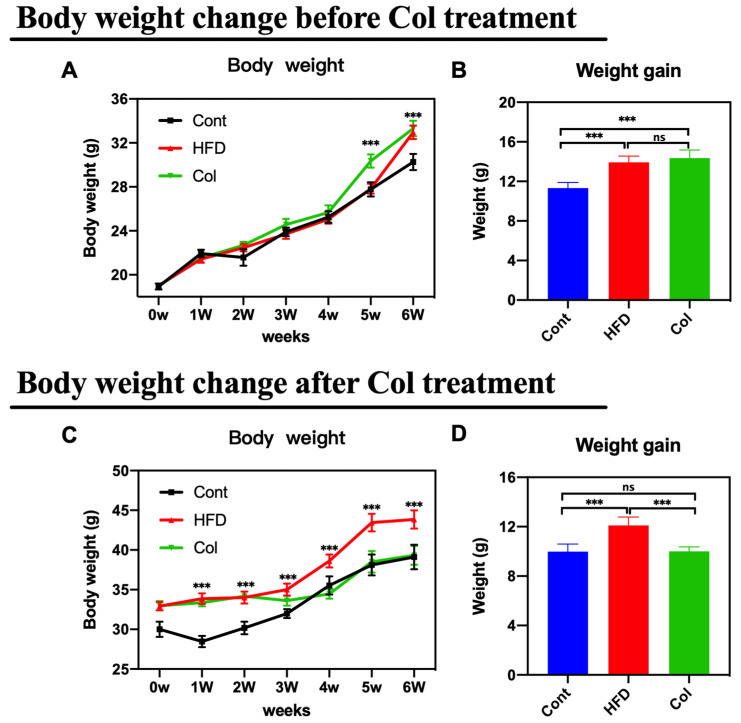
Canola oil inhibits the body weight gain of obese mice. (**A**) Body weight change during the no-treatment experiment stage (high-fat diet-induced obesity stage); (**B**) body weight gain during the no-treatment experiment stage; (**C**) body weight change during the Col-treatment experiment stage; (**D**) body weight gain during the Col-treatment experiment stage. Data are expressed as the mean ± SEM (*n* = 10), * *p* < 0.05, ** *p* < 0.01, *** *p* < 0.001, and ns *p* > 0.05.

**Figure 2 nutrients-16-03379-f002:**
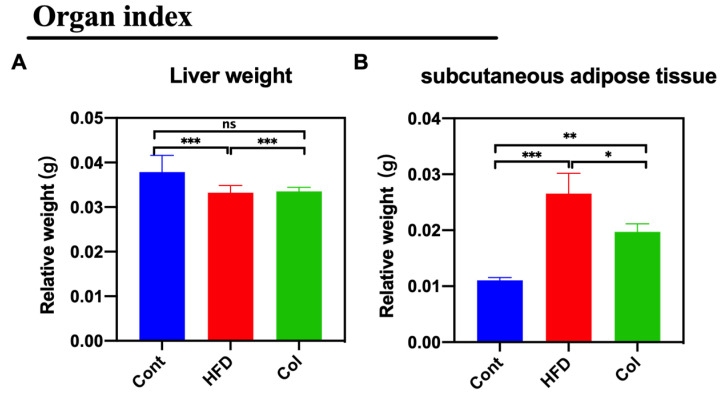
Canola oil inhibits the subcutaneous adipose tissue accumulation in obese mice. (**A**) Relative weight of liver to body weight; (**B**) relative weight of subcutaneous adipose tissue to body weight. Data are expressed as the mean ± SEM (*n* = 10), * *p* < 0.05, ***p* < 0.01, ****p* < 0.001, and ns *p* > 0.05.

**Figure 3 nutrients-16-03379-f003:**
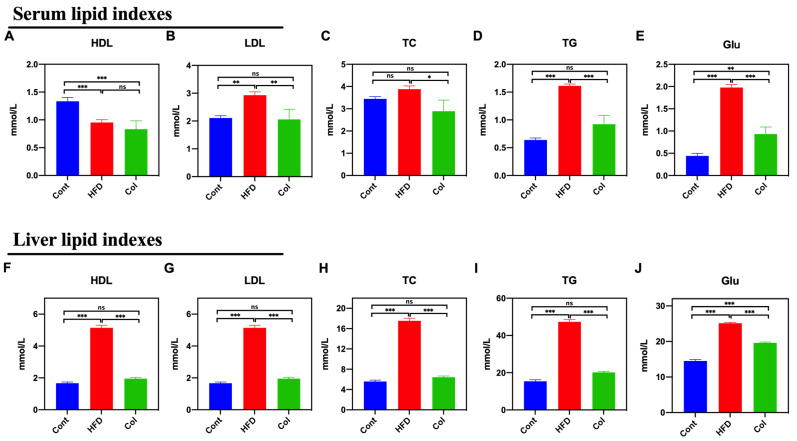
Canola oil inhibits the biochemical indexes of lipids in obese mice. Serum HDL (**A**), LDL (**B**), TC (**C**), TG (**D**), and Glu (**E**) levels in serum; HDL (**F**), LDL (**G**), TC (**H**), TG (**I**), and Glu (**J**) in liver. Data are expressed as the mean ± SEM (*n* = 10), * *p* < 0.05, ** *p* < 0.01, *** *p* < 0.001, and ns *p* > 0.05.

**Figure 4 nutrients-16-03379-f004:**
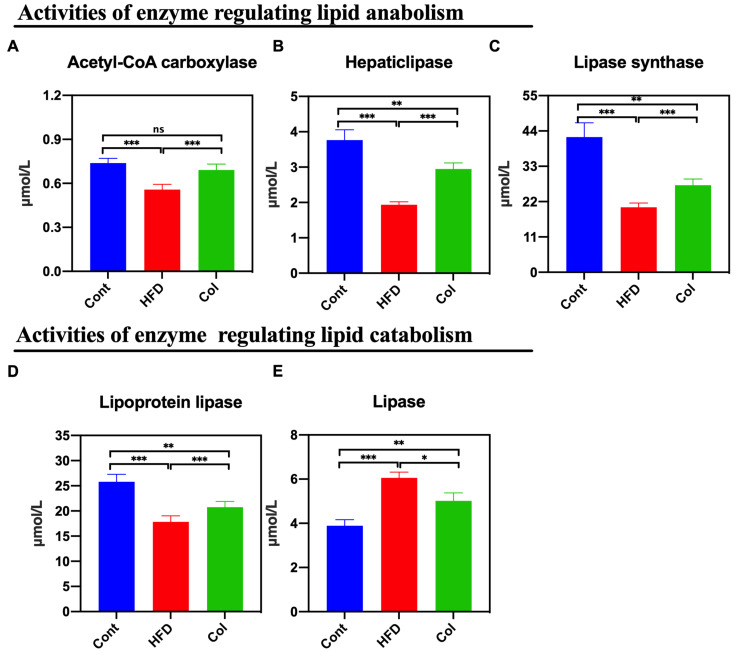
Canola oil regulates the hepatic lipase bioactivity of obese mice. (**A**) Activity of acetyl-CoA carboxylase; (**B**) hepatic lipase; (**C**) lipase synthase; (**D**) lipoprotein lipase; (**E**) lipase. Data are expressed as the mean ± SEM (*n* = 10), * *p* < 0.05, ** *p* < 0.01, *** *p* < 0.001, and ns *p* > 0.05.

**Figure 5 nutrients-16-03379-f005:**
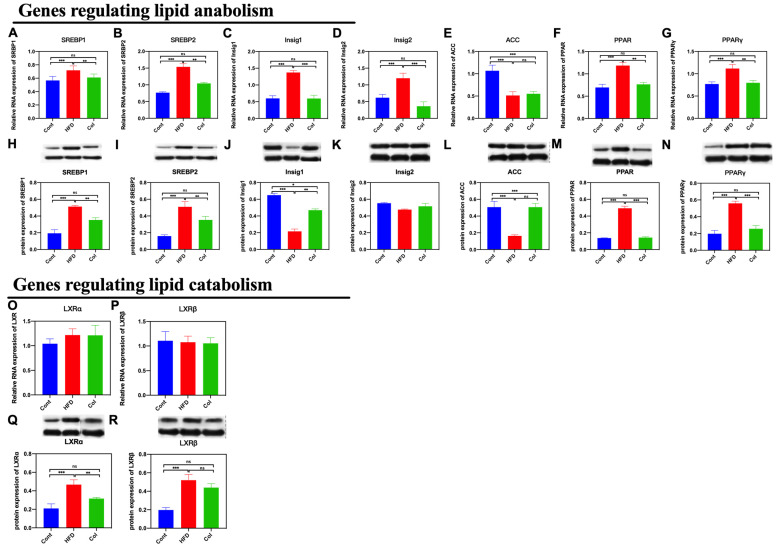
Canola oil inhibits the hepatic lipid anabolism of obese mice. (**A**,**B**,**H**,**I**) mRNA and protein expression of SREBP1/2; (**C**,**D**,**J**,**K**) mRNA and protein expression of Insig1/2; (**E**,**L**) mRNA and protein expression of ACC; (**F**,**G**,**M**,**N**) mRNA and protein expression of PPAR/PPARγ; (**O**–**R**) mRNA and protein expression of LXRα/LXRβ. Data are expressed as the mean ± SEM (*n* = 10), * *p* < 0.05, ** *p* < 0.01, *** *p* < 0.001, and ns *p* > 0.05.

**Figure 6 nutrients-16-03379-f006:**
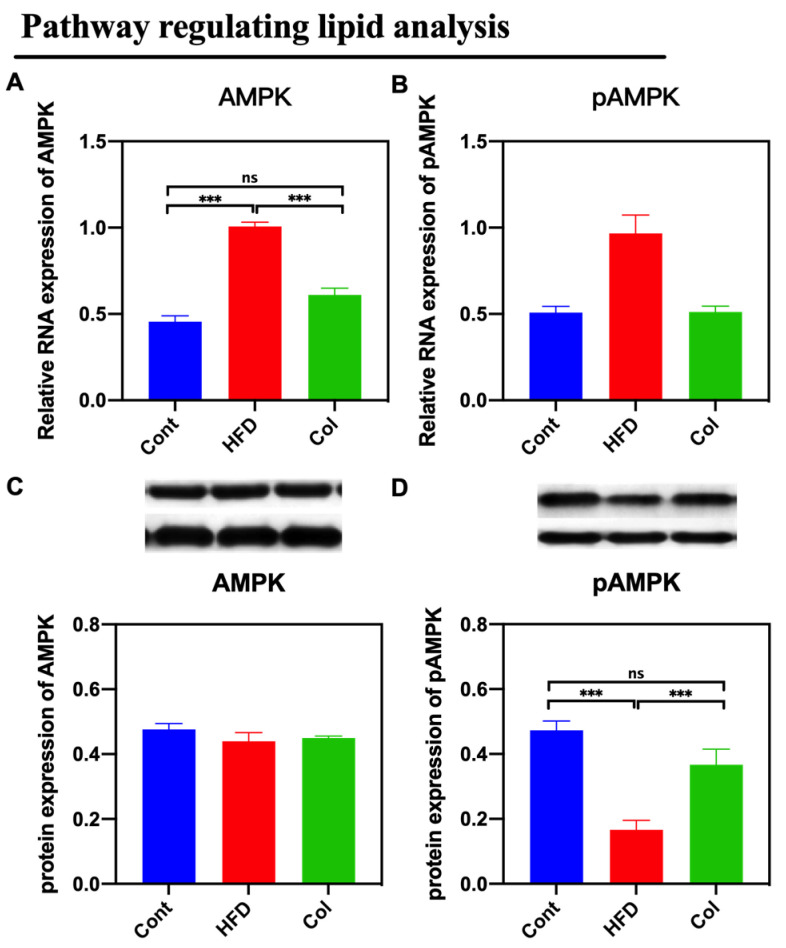
Canola oil inhibits the AMPK pathway activity of obese mice. (**A**) mRNA expression of AMPK; (**B**) mRNA expression of pAMPK. (**C**) Protein expression of pAMPK; (**D**) Protein expression of pAMPK. Data are expressed as the mean ± SEM (*n* = 10), * *p* < 0.05, ** *p* < 0.01, *** *p* < 0.001, and ns *p* > 0.05.

**Figure 7 nutrients-16-03379-f007:**
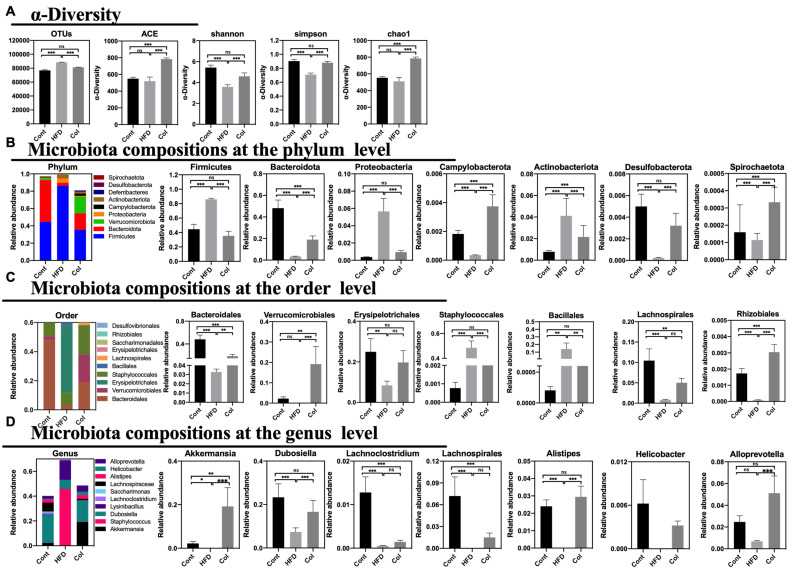
Canola oil reverses dysbiosis with respect to the composition and relative abundance of the gut microbiota. (**A**) OTU annotation of fecal microbiota and the alpha-diversity of fecal microbiota; (**B**) composition of microbiota at the phylum level; (**C**) composition of microbiota at the order level; (**D**) composition of microbiota at the genus level. Data are expressed as the mean ± SEM (*n* = 10), * *p* < 0.05, ** *p* < 0.01, *** *p* < 0.001, and ns *p* > 0.05.

**Figure 8 nutrients-16-03379-f008:**
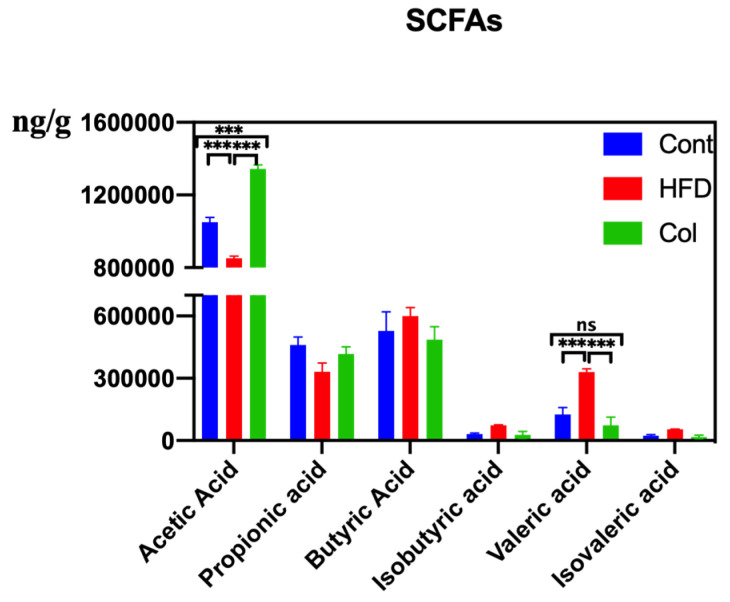
Canola oil influences the SCFAs concentration of obese mice. Data are expressed as the mean ± SEM (*n* = 10), * *p* < 0.05, ** *p* < 0.01, *** *p* < 0.001, and ns *p* > 0.05.

**Figure 9 nutrients-16-03379-f009:**
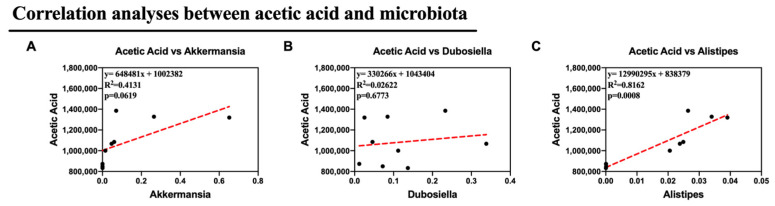
Col may exhibit the anti-obesity roles via microbiota-mediated acetic acid of obese mice (**A**) The correlation analyses between *Akkermansia* and acetic acid content; (**B**) The correlation analyses between *Dubosiella* and acetic acid content; (**C**) The correlation analyses between *Alistipes* and acetic acid content. Dot: samples in group; red dash line: the correlation analyses line. Data are expressed as the mean ± SEM (*n* = 10), * *p* < 0.05, ** *p* < 0.01, *** *p* < 0.001, and ns *p* > 0.05.

**Figure 10 nutrients-16-03379-f010:**
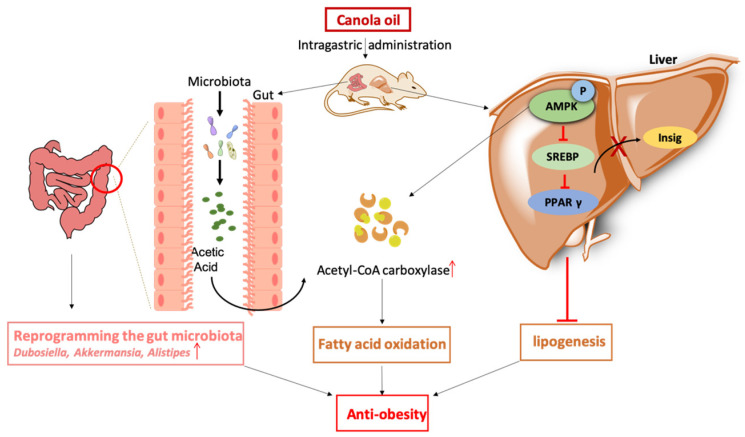
Col ameliorated obesity and suppressed lipogenesis in HFD mice. The underlying mechanisms are possibly associated with the reprogramming of the gut microbiota, in particular, the acetic acid-mediated increased expression of *Alistipes* via the AMPK signaling pathway. ↑—increased expression.

**Table 1 nutrients-16-03379-t001:** The relative content of canola oil.

Fatty Acid	Relative Content
Myristic Acid (C14:0)	0.05%
Palmitic Acid (C16:0)	4.37%
Palmitoleic Acid (C16:1n7)	0.28%
Heptadecanoic Acid (C17:0)	1.64%
10-Heptadecenoicacid (C17:1n7)	0.08%
Stearic Acid (C18:0)	1.81%
Oleic Acid (C18:1n9c)	60.60%
Linoleic Acid (C18:2n6c)	19.80%
Linolenic Acid (C18:3n3)	8.96%
Arachidic Acid (C20:0)	0.61%
11-Eicosenoic Acid (C20:1)	0.37%

**Table 2 nutrients-16-03379-t002:** Quantitative PCR primers used in this study.

Gene	Forward (5′-3′)	Reverse (5′-3′)
PPAR	AGGCTGTAAGGGCTTCTTTCG	GGCATTTGTTCCGGTTCTTC
PPARγ	CCATTCTGGCCCACCAAC	AATGCGAGTGGTCTTCCATCA
SREBP1	GAACGACATCGAAGACATGC	GAGAAGCTCTCAGGAGAG
SREBP2	GTGCGCTCTCGTTTTACTGAAGT	GTATAGAAGACGGCCTTCACCAA
LXRɑ	CTCAATGCCTGATGTTTCTCCT	TCCAACCCTATCCCTAAAGCAA
LXRβ	GATCCTCCTCCAGGCTCTGAA	TGCGCTCAGGCTCATCCT
ACC	TGGGTTGAGGAATGTGTTGGTG	GAAGTAGCCGATGAGGATGATG
Insig1	ATCACGCCAGTGCTAAAGTA	CAACCAAGAACGGACATAGA
Insig2	ATCACGCCAGTGCTAAAGTA	CAACCAAGAACGGACATAGA
β-actin	GTCCACCTTCCAGCAGATGT	GAAAGGGTGTAAAACGCAGC

## Data Availability

The original contributions presented in the study are included in the article, further inquiries can be directed to the corresponding author.

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
