# Peer review of "Canola Oil Ameliorates Obesity by Suppressing Lipogenesis and Reprogramming the Gut Microbiota in Mice via the AMPK Pathway"

_nutrients, 2024, doi:10.3390/nu16193379_

Round 1

Reviewer 1 Report

Comments and Suggestions for Authors

1.I think this is an excellent paper with numerous related experiments conducted.

2.It is known that gut microbiota are regulated by Col (choline), but if there are known results regarding its effects on the gastrointestinal tract, please supplement the explanation.

3.Various ion channels are involved in the regulation of obesity, but is the regulation of ion channels by Col well known? If so, please provide additional explanations.

4.How can Col be applied clinically to humans?

5.The mechanism of lipid metabolism regulation has been presented in relation to AMPK, but what about the involvement of other pathways like MAPK?

Author Response

Journal: Nutrients

Manuscript Number: nutrients-3228218

Referee: 1

Comments to the Author

Title: Camellia (Camellia oleifera bel.) seed oil reprogramming of gut microbiota and Alleviates Lipid Accumulation in high fat-fed mice through mTOR pathway

1.I think this is an excellent paper with numerous related experiments conducted.

Answer: Thank you very much for your professional endorsement.

2.It is known that gut microbiota are regulated by Col (choline), but if there are known results regarding its effects on the gastrointestinal tract, please supplement the explanation.

Answer: According to previous study, dysregulation of choline also contributes to lipid accumulation and to a chronic inflammatory status. (Chen J, Vitetta L. Gut Microbiota Metabolites in NAFLD Pathogenesis and Therapeutic Implications. Int J Mol Sci. 2020 Jul 23;21(15):5214. doi: 10.3390/ijms21155214. PMID: 32717871; PMCID: PMC7432372). Regrettably, this current trial doesn't cover the relationship between choline and gastrointestinal tract, I believe that the outcomes will be meaningful.

3.Various ion channels are involved in the regulation of obesity, but is the regulation of ion channels by Col well known? If so, please provide additional explanations.

Answer: Thank you for your comments, the regulation of ion channels by Col could found in reference 41. According to Chen, L et.al, Col is the third most consumed culinary oil in the world. It is well-known for its high content of unsaturated fatty acids, especially polyunsaturated fatty acids, which make it of great nutritional value. There is increasing evidence that a diet rich in unsaturated fatty acids offers health benefits. rapeseed oil that contributes to its anti-microbial, anti-inflammatory, anti-obesity, anti-diabetic, anti-cancer, neuroprotective, and cardioprotective, among others.

4.How can Col be applied clinically to humans?

Answer: According to studies found in PubMed, Col has been used clinically for Alzheimer's disease, a variety of cardiovascular diseases ,nutritional disorders and other diseases.

5.The mechanism of lipid metabolism regulation has been presented in relation to AMPK, but what about the involvement of other pathways like MAPK?

Answer: Thank you for your sincere suggestions, I have made some revisions to the manuscript based on your comments.

Reviewer 2 Report

Comments and Suggestions for Authors

Thank you very much for your interesting research.

Some points must be carefully revised:

General. Please, revise the typos and grammar throughout the whole manuscript.

Abstract. Abbreviations should be described and added in brackets in their first inclusion in the text (for instance, TC and TG).

Introduction. Line 53. Please avoid the use of first-person pronouns. Also in line 87. Please, revise this issue throughout the whole text.

Materials and Methods. Line 96. Although a previous article is cited, please, include some information about this methodology.

Materials and Methods. Table 1. This table seems to be a Result and should be placed in the Results section (in my opinion).

Materials and Methods. Protease activity assay: description of the method?

Results. Lines 243-245. References related to these mentioned results must be added.

Results. Figure 4E. Labels for X axis seem to be incorrect.

Conclusion. Current limitations and future perspectives must be included.

Author Response

Journal: Nutrients

Manuscript Number: nutrients-3228218

Comments to the Author

Title: Canola Oil Ameliorates Obesity by Suppressing Lipogenesis and Reprogramming Gut Microbiota in mice via the AMPK pathway

Referee: 2

Comments to the Author

General. Please, revise the typos and grammar throughout the whole manuscript.

Answer: Thank you for your sincere suggestions, I have made some revisions to the manuscript based on your comments.

Abstract. Abbreviations should be described and added in brackets in their first inclusion in the text (for instance, TC and TG).

Answer: Appropriate adjustments were made based on your comments.

Introduction. Line 53. Please avoid the use of first-person pronouns. Also in line 87. Please, revise this issue throughout the whole text.

Answer: Thank you for your sincere suggestions, I have made some revisions throughout the whole textbased on your comments.

Materials and Methods. Line 96. Although a previous article is cited, please, include some information about this methodology.

Answer: Modifications in line 97-102 have been made according to your suggestions.

Materials and Methods. Table 1. This table seems to be a Result and should be placed in the Results section (in my opinion).

Answer: Thank you for your sincere suggestions, as the content of table1 was not included as a separate result, so it was put in the materials and methods section.

Materials and Methods. Protease activity assay: description of the method?

Answer: Thank you for your comments, the description has been added in line 143-146.

Results. Lines 243-245. References related to these mentioned results must be added.

Answer: References related to these mentioned results in line 253 were added based on your comments.

Results. Figure 4E. Labels for X axis seem to be incorrect.

Answer: Thank you for your carefully review. It was my negligence. I've adjusted the mistake in lines 258-273.

Round 2

Reviewer 1 Report

Comments and Suggestions for Authors

It is well revised.

Reviewer 2 Report

Comments and Suggestions for Authors

Thank you for addressing all the comments and submitting this improved version.